# Implications of Senescent T Cells for Cancer Immunotherapy

**DOI:** 10.3390/cancers15245835

**Published:** 2023-12-14

**Authors:** Tetsuhiro Kasamatsu

**Affiliations:** Department of Laboratory Sciences, Gunma University Graduate School of Health Sciences, 3-39-22 Showa-machi, Maebashi 371-8514, Gunma, Japan; kasamatsu@gunma-u.ac.jp; Tel.: +81-27-220-8994

**Keywords:** therapy-induced cellular senescence, T-cell senescence, PD-1

## Abstract

**Simple Summary:**

Senescent T cells are defective in proliferation and effector functions, and they accumulate owing to aging, chronic viral infections, and autoimmune diseases. Several DNA-damaging chemotherapeutic agents, such as melphalan and doxorubicin, induce senescence in cancer cells. However, it is unclear whether these agents induce senescence in T cells. We evaluated whether sublethal doses of melphalan and doxorubicin induce cellular senescence in human peripheral blood mononuclear cell-derived T cells from healthy donors. Our results demonstrate that therapy-induced cellular senescence is also induced in T cells and that PD-1 may be a biomarker for therapy-induced senescent T cells. In this review, we discuss the characteristics of senescent T-cells, including therapy-induced senescent T-cells. In addition, we investigate whether therapy-induced cellular senescence has positive and/or negative effects on T cells.

**Abstract:**

T-cell senescence is thought to result from the age-related loss of the ability to mount effective responses to pathogens and tumor cells. In addition to aging, T-cell senescence is caused by repeated antigenic stimulation and chronic inflammation. Moreover, we demonstrated that T-cell senescence was induced by treatment with DNA-damaging chemotherapeutic agents. The characteristics of therapy-induced senescent T (TIS-T) cells and general senescent T cells are largely similar. Senescent T cells demonstrate an increase in the senescence-associated beta-galactosidase-positive population, cell cycle arrest, secretion of senescence-associated secretory phenotypic factors, and metabolic reprogramming. Furthermore, senescent T cells downregulate the expression of the co-stimulatory molecules CD27 and CD28 and upregulate natural killer cell-related molecules. Moreover, TIS-T cells showed increased PD-1 expression. However, the loss of proliferative capacity and decreased expression of co-stimulatory molecules associated with T-cell senescence cause a decrease in T-cell immunocompetence. In this review, we discuss the characteristics of senescent T-cells, including therapy-induced senescent T cells.

## 1. Introduction

T cells are a subtype of white blood cells that play a key role in anticancer immunity. Cytotoxic T cells have long been recognized as immunological effectors that mediate tumor eradication. Helper T cells provide the requisite pro-inflammatory cytokines and chemokines to recruit other immune cells and maintain effective cellular immune responses. Immunotherapy is a type of treatment that helps the immune system combat cancer. Recently, a growing interest has been observed in new cancer immunotherapies aimed at inducing T cell-mediated anti-tumor responses. These include immune checkpoint inhibitors (ICIs) and Chimeric Antigen Receptor (CAR)-T cells. These immunotherapies also enhance the ability of the T-cell immune system to target cancer cells. Therefore, the efficacy of these treatments depends on the condition of the patient’s immune system.

T-cell senescence is assumed to be attributed to the age-dependent loss of the ability to m T cells, which are a white blood cell subtype that plays a key role in anticancer immunity. Cytotoxic T cells have long been recognized as immunological effectors that mediate tumor eradication. Helper T cells provide the requisite proinflammatory cytokines and chemokines to recruit other immune cells and maintain effective cellular immune responses. T-cell receptors (TCRs) are protein complexes on the T cell surface that recognize antigen fragments as peptides bound to major histocompatibility complex (MHC) molecules. TCRs are composed of a combination of TCRα and -β chains [1,2]. Through a process called somatic V(D)–J recombination, the TCR gene gives rise to a diverse T-cell repertoire that can recognize a wide range of antigens [3]. Upon binding to a specific peptide loaded onto an MHC molecule, the TCR initiates a signaling cascade involving the activation of transcription factors and cytoskeletal reorganization, resulting in T-cell activation. Activated T cells secrete cytokines, proliferate rapidly, exhibit cytotoxic activity, and differentiate into effector and memory cells. Immunotherapy is a type of treatment that helps the immune system combat cancer. Recently, there has been growing interest in new cancer immunotherapies aimed at inducing T cell-mediated anti-tumor responses. These include immune checkpoint inhibitors (ICIs) and chimeric antigen receptor (CAR)-T cells. Immunotherapies also enhance the ability of the T-cell immune system to target cancer cells. Therefore, the efficacy of these treatments depends on the immune system of the patient.

Cellular senescence is an arrest of the cell cycle caused by internal factors (oxidative damage, telomere loss, and hyperproliferation) and/or external factors (UV light, γ-irradiation, chemotherapeutic drugs) in response to DNA damage (DDR) [4]. Cellular senescence is thought to exert beneficial effects in the early stages by halting cancer development and promoting survival; however, it is proposed to have detrimental effects in the later stages, as senescent cells accumulate due to aging and improper removal [5,6]. The expression of PD-1 ligand-1 (PD-L1, CD274), an immune checkpoint molecule, is enhanced in senescent cells, and ICI administration improves the senescence-associated phenotype [7]. In addition, CAR-T cells targeting urokinase-type plasminogen activator receptor (uPAR) and dipeptidyl peptidase 4 (DPP4), which are expressed in senescent cells, have been shown to reduce the number of senescent cells [8,9]. These findings suggest the effectiveness of cancer immunotherapy for targeting senescent cells.

In contrast, T-cell senescence is assumed to be attributable to the age-dependent loss of the ability to mount an effective response against pathogens and tumor cells. However, an increase in senescent T cells has also been detected in young patients with X-linked lymphoproliferative syndrome after primary Epstein–Barr virus infection [10]. In addition to aging, T-cell senescence is caused by repeated antigenic stimulation and chronic inflammation. Senescent T cells have also been detected in the peripheral blood of patients with hematologic malignancies, such as acute myeloid leukemia (AML) and solid tumors [11,12,13,14]. The direct induction of T-cell senescence by tumor cells has been observed in vitro [15].

Melphalan (MEL) and doxorubicin (DXR) are chemotherapeutic agents that are widely used to treat various cancers. These chemotherapeutic agents can induce deoxyribonucleic acid (DNA) double-stranded breaks (DSBs) [16], which result in the phosphorylation of histone H2AX by activating ataxia-telangiectasia-mutated (ATM) kinases. Phosphorylated histone H2AX (γH2AX) recruits DNA damage response proteins [17]. When the DNA damage is severe, cells initiate apoptosis. In contrast, cells initiate cellular senescence when DNA damage is relatively mild [18]. In many cases, patients receive DNA-damaging chemotherapeutic agents before immunotherapy. We hypothesized that the induction of T-cell senescence by these drugs would affect the therapeutic effect of immunotherapy. We have previously demonstrated that sublethal doses of chemotherapeutic agents induce cellular senescence in T cells derived from healthy donor human peripheral blood mononuclear cells (PBMNCs) [19].

In this review, we discuss the characteristics of senescent T cells, including therapy-induced senescent T (TIS-T) cells. In addition, we investigate whether therapy-induced cellular senescence has positive and/or negative effects on T cells.

## 2. Functional Markers

### 2.1. Senescent T Cell Characteristics

T-cell senescence shares several characteristics with somatic cell senescence. In previous studies, senescent T cells also demonstrated an increase in the senescence-associated beta-galactosidase (SA-β-Gal)-positive cell population [20,21] and increased DNA damage, cell cycle arrest [22,23], secretion of senescence-associated secretory phenotype (SASP) factors [20,24], and metabolic reprogramming [23,25].

Mitogen-activated protein kinases (MAPKs) such as extracellular signal-regulated protein kinases (ERKs), c-Jun N-terminal kinase (JNK), and p38 play a critical role in regulating T-cell senescence. In senescent T cells, MAPKs are activated in response to metabolic disturbances, such as DNA damage [26], glucose consumption [26], and cyclic adenosine monophosphate (cAMP) accumulation [27]. The activation of JNK, ERK, and p38 suppresses T cell proliferation [26,28]. Activated p38 and ERK, in cooperation with signal transducer and activator of transcription 1/3 (STAT1/3), markedly increase the expression of cyclin-dependent kinase inhibitors (CKI), such as p21, p16, and p53, and prevent T cell proliferation [23]. In mammalian cells, ataxia-telangiectasia mutated (ATM), ATM- and Rad3-related (ATR), and DNA-dependent protein kinase (DNA-PK) are the most upstream DNA damage-response signaling pathways. ATM and ATR also activate a second wave of phosphorylation through the activation of checkpoint kinase 1 (Chk1), checkpoint kinase 2 (Chk2), and MAP kinase-activated protein kinase 2 (MK2) and the activation of p53 [29]. ATM-associated DNA damage and cell senescence are regulated by MAPK, ERK1/2, and p38 signaling [30].

In a study using single-cell RNA sequencing (sc-RNA-seq), antioxidant factors including superoxidedismutase1 (Sod1) and nuclear factor erythroid 2–related factor 2 (Nrf2) were found to be expressed in senescent T cells in mice [31]. Similarly, an increased expression in Sestrin 3 (SESN3), which is stimulated by the oxidative stress response to forkhead box O3 (FOXO3), has been observed in human senescent T cells using scRNA-seq [32]. In addition, serine/threonine protein kinase 17A (STK17A) is associated with senescent T cells [32]. STK17A is involved in DNA damage response, positive regulation of apoptosis, and mitochondrial and metabolic regulation of reactive oxygen species (ROS).

### 2.2. Therapy-Induced Senescent T Cells

The characteristics of TIS-T cells are common to those of general T-cell senescence. In our previous study, sublethal doses of chemotherapeutic agents resulted in similar senescence-related features in T cells [19].

## 3. Cell Surface Markers (Figure 1)

### 3.1. Senescent T Cell Characteristics

Costimulatory molecules are a heterogeneous group of cell surface molecules that amplify or counteract the initial activation signal provided by TCRs to T cells following their interaction with the antigen/MHC. Therefore, costimulatory molecules affect T cell differentiation and fate. Furthermore, senescent T cells not only downregulate the expression of the co-stimulatory molecules CD27 and CD28, but also upregulate the expression of natural killer (NK) cell-related molecules such as CD57 and killer cell lectin-like receptor subfamily G member 1 (KLRG-1) [33]. Downregulation of co-stimulatory molecules and upregulation of NK cell-related molecules have also been confirmed in the peripheral blood of patients with solid cancers, as well as those with hematological malignancies such as lung cancer, breast cancer, gastric cancer, and AML [11,12,13,14]. Some studies have demonstrated that repeated cytotoxic chemotherapy induces T-cell senescence in the peripheral blood of patients [34,35]. In patients with diffuse large B-cell lymphoma (DLBCL), the loss of CD27 and CD28 expression in T cells correlates with the number of prior chemotherapy cycles [34]. In patients with multiple myeloma, high-dose MEL followed by autologous stem cell transplantation increases the co-stimulated molecular double-negative fraction of T cells [35].

**Figure 1 cancers-15-05835-f001:**
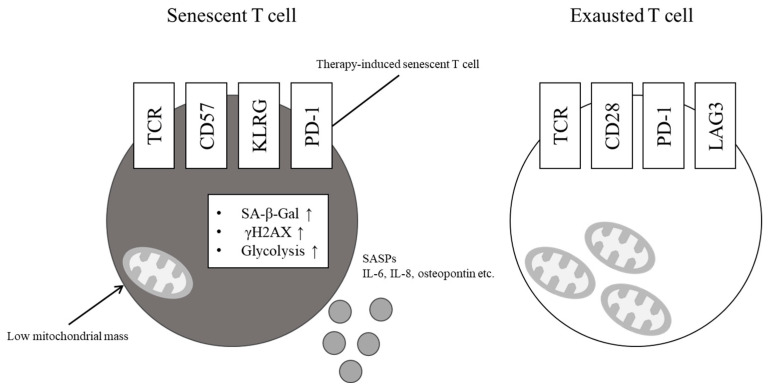
Characteristics of senescent T cell and exhausted T cell. Senescent T cells demonstrate increased senescence-associated beta-galactosidase (SA-β-Gal) -positive population and phosphorylated histone H2AX (γH2AX) nuclear foci. The expression of CD57 and killer cell lectin-like receptor subfamily G member 1 (KLRG-1) is upregulated, while the co-stimulatory molecules CD27 and CD28 are downregulated in senescent T cells. Senescent T cells have defective mitochondria with a low mitochondrial mass and are heavily dependent on glycolysis. Senescence-associated secretory phenotype (SASP) factors, such as interleukin (IL)-6, IL-8 and osteopontin, are upregulated.

### 3.2. Therapy-Induced Senescent T Cells

We examined whether sublethal doses of MEL and DXR induced cellular senescence in T cells from human peripheral blood mononuclear cells from healthy donors [19] and found that PDCD1 messenger ribonucleic acid (mRNA) and cell surface programmed death 1 (PD-1, also known as PDCD1) expression in T cells were upregulated by sublethal doses of chemotherapeutic agents. Additionally, PD-1 plays a central role in immune checkpoint pathways. Immune checkpoint pathways are critical modulators of the immune system that allow the initiation of the immune response and prevent the onset of autoimmunity. Moreover, PD-1 is expressed on activated T cells and suppresses the activation and cytokine production of lymphocytes by interacting with its ligands, PD-L1 and PD-1 ligand-2 (PD-L2 and PDCD1LG2) [36,37]. In senescent T cells, PD-1 expression remains controversial. PD-1 is a well-known marker of exhausted T cells [38]. In contrast, Janelle et al. demonstrated a strong relationship between senescence features and PD-1 expression in humans and mice [39]. The most widely studied pathways regulating cellular senescence caused by DNA damage are p16INK4a/retinoblastoma (Rb) and/or p53/p21Cip1 [40]. Janelle et al. demonstrated that the development of cellular senescence features in PD-1+ T cells is caused by the upregulation of p16INK4a via the p38MAPK signaling pathway [39]. Moreover, Muroyama et al. reported that the expression of phosphorylated ATM induced by the DNA damage response was strongly correlated with PD-1 expression in CD8+ T cells [41]. Based on these reports, elevated PD-1 expression may be specific to TIS-T cells.

### 3.3. Senescent T Cells and Exhausted T Cells

T cell exhaustion and senescence are two dominant dysfunctional states that differ phenotypically and functionally from other T cell states under certain pathological conditions, including chronic infections and cancers. Exhausted and senescent T cells have several overlapping phenotypic and functional features, including defective proliferative activity, reduced cytotoxic activity, and cell cycle arrest [42,43,44]. However, each state has unique molecular characteristics, including surface molecules, cytokines, and transcription profiles.

T-cell exhaustion is another dysfunctional condition observed in chronic infections and cancers. This condition is characterized by poor effector function and sustained expression of inhibitory receptors, such as PD-1, cytotoxic T-lymphocyte-associated protein 4, and lymphocyte activation gene-3 (LAG-3) [42]. Other characteristics of exhausted T cells include reduced cytotoxicity and production of cytokines, including IL-2, tumor necrosis factor (TNF) -alpha, and interferon (IFN)-gamma [42]. Transcriptional regulation controls T cell exhaustion. The transcription factors’ nuclear factor of activated T cells (NFAT), nuclear receptor 4a (Nr4a), and thymocyte selection associated with the high mobility group box (TOX), specifically activate an exhaustion-related transcriptional program that causes T-cell exhaustion [45].

PD-1 blockade increases the function of exhausted CD8+ T cells [46]. Thus, the exhausted state of CD8+ T cells is reversible [47]. On the other hand, “true” senescent cells are irreversible. However, the irreversibility of senescent T-cells has been questioned. Recent studies have shown that senescent T cells can regain function by inhibiting the p38 mitogen-activated protein kinase (MAPK) pathway [26].

## 4. Senescence-Associated Secretory Phenotype

### 4.1. Senescent T Cell Characteristics

Senescent cells produce proinflammatory and matrix-degrading molecules known as the senescence-associated secretory phenotype (SASP). This could be attributed to the elimination of senescent cells by immune cells, which are attracted to pro-inflammatory and chemotactic factors released as part of the SASP [48]. Another important physiological role of the SASP is the repair of damaged tissue. Campisi et al. reported the transient production of senescent cells by the SASP in subcutaneous fibroblasts [49]. Senescent T cells also produce inflammatory cytokines and chemokines that are termed SASP factors. SASP factors, such as interleukin (IL-6, IL-8, interferon-gamma, TNF-alpha, C-X-C motif chemokine receptor (CXCR) 1, and CXCR2, have been reported to be significantly upregulated in senescent cells [20,50,51]. However, senescent T cells are defective in the production of regular T cell cytokines such as IL-2 and IL-4 [24,52]. Additionally, IL-6, a multidirectional inflammatory cytokine, is the most prominent cytokine in the SASP and has been associated with DNA damage and oncogenic stress-induced senescence in keratinocytes, melanocytes, monocytes, fibroblasts, and epithelial cells in mice and humans [53]. Senescent T cells from patients with lung cancer display elevated mRNA expression of IL-6 and IL-8 mRNA expression [50]. The secreted phosphoprotein 1 (SPP1) gene encodes SPP1 (also known as osteopontin [OPN]), a potent inflammatory cytokine secreted by many cell types. In addition, SPP1 has been reported as a marker of age-dependent T-cell senescence [54].

### 4.2. Therapy-Induced Senescent T Cells

Breast cancer patients treated with DXR and cyclophosphamide show an increased expression in p16INK4a in T cells and increased serum levels of senescence-associated cytokines (vascular endothelial growth factor A and monocyte chemotactic protein 1). However, studies directly evaluating SASP production by treatment-induced senescent T cells are lacking. In our study, IL-6 and SPP1 mRNA expression in T cells was upregulated by sublethal doses of chemotherapeutic agents [19].

## 5. Differentiation Phenotypes

During differentiation, selection, and proliferation, T cell development results in T cells ready for circulation within the peripheral blood. These inactivated T cells are naïve. Upon encountering the antigen, naïve T cells expand clonally and differentiate into effector T cells, known as T helper (TH) cells and cytotoxic T lymphocytes (CTL) for CD4+ and CD8+ T cells, respectively. After antigen-induced proliferation and death of effector cells following antigen clearance, some of the remaining T cells differentiate into two distinct types of memory T cells: central memory T (TCM) cells (CD45RO+/CCR7+) and effector memory T (TEM) (CD45RO +/ CCR7−) cells. Furthermore, some TEM cells that are CD27−/CD28− re-express cell surface CD45RA and are known as the terminal effector memory T (TEMRA) cells (CD45RA+/CCR7−) subset [55]. TEMRA cells exhibit many features that are common to cellular senescence, including decreased proliferation, defective mitochondrial function, and elevated p38 MAPK signaling [56].

Lee et al. demonstrated that tumor-infiltrating TEMRA cells differ greatly from peripheral blood TEMRA cells with distinct transcriptomes and functional properties [57]. The majority of tumor-infiltrating TEMRA cells were CD27+/CD28+ double-positive, which is a characteristic of poorly differentiated effector cells [57]. In contrast, the majority of TEMRA cells in the peripheral blood were CD27−/CD28− double negative. Moreover, the proportion of CD27+/CD28 TEMRA cells is inversely correlated with the responsiveness to ICI therapy [57]. Using single-cell RNA sequencing, transcription factors associated with TEMRA cells, such as the T-box transcription factor 21, have been correlated with TCR clonality in various cancers [58].

Epigenetic changes are associated with T-cell differentiation. DNA methylation studies have demonstrated that senescence of the human immune system is associated with methylation changes at specific CpG sites [59,60,61]. Naïve and central memory T cells from older individuals exhibit a shift toward more differentiated patterns of chromatin openness compared to young individuals [62]. Using an assay for transposase-accessible chromatin with high-throughput sequencing, Ucar et al. found that several histone genes (HIST1H3D, HIST1H3E, and HIST4H4) are closely related to chromatin during aging, consistent with decreased expression of core histones in model systems of aging [63]. Furthermore, ten-eleven translocation 2 (TET2), which requires α-ketoglutarate as a catalytic substrate to convert methyl cytosine to 5-hydroxymethylcytosine, is rapidly upregulated upon TCR signaling activation. Loss of TET2 shifts CD8+ T cell differentiation toward central memory phenotypes and promotes secondary recall responses [64].

## 6. Immunosuppression

These cells are unable to respond to stimulation or recognition of tumor antigens because of the downregulation of the co-stimulatory molecules CD27 and CD28 and the upregulation of inhibitory molecules, including LAG-3 and PD-1 [20,33,42]. In addition, the decreased production of perforin and granzyme impairs the effector function of T cells [65,66]. Senescent T cells may also enhance the production of inflammatory cytokines (TNF, IL-1β, and IL-6) and angiogenic factors (matrix metalloprotein 9, IL-8) by monocytes/macrophages and promote tubule formation and tumor cell survival [67]. Increased numbers of tumor cells may further induce T-cell senescence, potentially leading to immune evasion by pathogens and tumors.

## 7. Metabolic Features

### 7.1. Senescent T Cell Characteristics

T cell proliferation and effector functions require energy. Senescent T cells have defective mitochondria, the most important energy-producing organelles, with low mitochondrial mass, reduced mitochondrial membrane potential, and elevated ROS levels [25,51,68]. Human CD4+ and CD8+ T cells differ in their susceptibility to senescence, with CD8+ T cells acquiring the senescent phenotype earlier than CD4+ T cells. Callender et al. demonstrated that inherent differences in mitochondrial content drive this phenotype, with senescent human CD8+ T cells displaying lower mitochondrial mass than CD4+ T cells [25]. Senescent CD4+ T cells consume more lipids and glucose than CD8+ T cells, leading to metabolic diversity involving either oxidative or glycolytic metabolism [25]. Moreover, mitochondrial dysfunction in human CD4+ T cells generates CD4+ T cells with a CD8+-like phenotype [25]. Mitochondrial mass modulates human T-cell senescence. Senescent T cells show impaired glucose transporter type 1 expression, resulting in impaired nutrient uptake [51]. Although effector T cells use glycolysis and oxidative phosphorylation to generate energy, senescent T cells are highly dependent on glycolysis. Autophagy has also been suggested as a source of nutrients during increased glycolysis [51].

Mitochondria are the primary source of reactive oxygen species in the cell and are byproducts of normal respiratory function. Under homeostatic conditions, mitochondrial ROS are produced at low concentrations and act as important signaling molecules involved in a wide variety of cellular processes, including immune responses [69]. Mild and low levels of ROS are essential for T cell activation and differentiation, whereas excessive ROS levels are closely linked to cellular senescence [69]. Overproduction of ROS has been observed in senescent T cells [51,70].

### 7.2. Therapy-Induced Senescent T Cells

Several antineoplastic agents have been shown to cause oxidative stress in patients undergoing cancer chemotherapy [71]. Agents that generate high levels of ROS include anthracyclines (e.g., DXR), alkylating agents (e.g., MEL), and platinum coordination complexes (e.g., cisplatin) [71]. Some chemotherapies increase ROS levels and promote T-cell senescence.

## 8. Positive or Negative Influence of Senescent T Cells (Table 1)

### 8.1. Gain of Innate-like Functions

TCRs can only recognize antigens bound to specific receptor molecules such as MHC I and MHC II. These MHC molecules are membrane-bound surface receptors on antigen-presenting cells such as dendritic cells and macrophages. To elicit an appropriate immune response, T cells must recognize foreign antigens bound to MHC molecules. Senescent T cells upregulate the expression of NK-like receptors, including KLRG-1 and NKG2A/C [72]. Jacomet et al. demonstrated that killer cell Ig-like receptor (KIR)/NKG2A+ T cells exert cytolytic activity in a TCR-independent manner [73]. Senescent T cells lose their antigen-specific killing capability but might preserve some anti-cancer immunity because of their robust nonspecific killing capacity.

### 8.2. Suppression of Anti-Tumor Immunity

However, the effector function of senescent T cells is reduced, and anti-tumor activity is not exhibited. Moreover, tumor-infiltrating T cells exhibit senescent characteristics at tumor sites in non-small cell lung cancer [11], multiple myeloma [74], ovarian cancer [72], breast cancer [72], and follicular lymphoma [75]. The direct induction of T-cell senescence by tumor cells has been observed in vitro [15]. Regulatory T (Treg) cells play a central role in regulating immune tolerance and homeostasis, preventing autoimmune diseases, and suppressing chronic inflammatory diseases [76]. However, Tregs can also inhibit effective immune responses against cancer [77]. High-Treg infiltration is correlated with poor survival in ovarian cancer [78], breast cancer [79], and hepatocellular carcinoma [80,81]. In vitro experiments have confirmed that Tregs in the tumor microenvironment can induce T-cell senescence [20]. Tumor cells or Tregs can directly induce T-cell senescence, thereby potentially utilizing senescent T-cells to evade anti-tumor immunity.

### 8.3. Prognostic Impact

Senescent T cells have been implicated in the prognosis of various tumors. The frequency of senescent T cells in patients with AML in complete remission (CR) is lower than the frequency of senescent T cells in patients with untreated AML [14]. Patients with stage IV lung cancer have an increased number of senescent T cells, which may be related to this disease [82]. In addition, the proportion of senescent T cells was highest 3 months after chemotherapy [82]. This suggests that chemotherapy induces T-cell senescence in vivo. Moreover, high levels of senescent T cells in the peripheral blood are associated with poor prognosis in gastric cancer [13], renal cell carcinoma [83], follicular lymphoma [75], and AML [84]. The detailed mechanisms underlying these effects are unclear; however, the induction and accumulation of senescent T cells as a result of repeated chemotherapy may be the contributing factors. Additionally, the abnormal secretion of inflammatory cytokines by SASP factors may be a contributing factor.

### 8.4. Effects of ICIs

Immunotherapy uses ICIs to treat melanoma, lung cancer, and other cancers. These drugs inhibit checkpoint proteins, including PD-1, PD-L1, and cytotoxic T-lymphocyte-associated protein-4. Anti-PD-1/PD-L1 monoclonal anti-bodies have been developed as ICIs to target immune evasion by tumors. Cancer cells undergo immune surveillance to enhance their survival and metastatic potential [85]. Anti-PD-1/PD-L1 monoclonal antibodies bind to PD-1 and PD-L1 receptors on T cells and tumor cells, respectively, disrupt the PD-1/PD-L1 interaction, and reactivate anti-tumor T cell responses. The anti-tumor effect of ICIs is associated with T-cell activation and CD28 expression [86]; therefore, therapy is only effective against exhausted T cells and not against senescent T cells. The loss of the surface markers CD27 and CD28 or the expression of Tim-3 and CD57 on T cells is associated with resistance to checkpoint inhibitor blockade in melanoma patients [87]. Treatment with a single anti-PD-1 antibody is effective [88]. This could be due to an increase in the number of aged T cells as well as exhausted T cells in the bone marrow of patients with myeloma [74]. In contrast, the prevention of tumor-specific T-cell senescence via ATM and/or MAPK signaling inhibition combined with anti-PD-L1 checkpoint blockade can synergistically enhance anti-tumor immunity and immunotherapy in vivo [89]. Classification of senescent and exhausted T cells using CD28 and CD57 may be useful for predicting ICI efficacy. However, differentiated CD4+ T cells do not upregulate CD57 and γH2AX despite exhibiting negative CD27 and CD28 levels [14]. Further studies are required to differentiate senescent T cells from exhausted T cells.

### 8.5. CAR-T Cell Therapies

CAR-T cell therapy is a cellular therapy that redirects a patient’s T cells to specifically target and destroy tumor cells. Moreover, CARs are engineered fusion proteins that genetically consist of an antigen-recognizing domain derived from a monoclonal antibody, and an intracellular T-cell signaling and costimulatory domain [90,91,92,93]. The first generation of CARs intracellular domains consists only of CD3ζ, while the second generation contains additional co-stimulatory signal transduction domains (CD28 or 4-1BB) [94]. The third generation combines two costimulatory domains (CD28 and 4-1BB) [94]. In 2017, the U.S. Food and Drug Administration approved the first two CD19-targeted CAR-T cell products. However, since 2021, two new products targeting B cell maturation antigens have become available for the treatment of relapsed/refractory multiple myeloma since 2021. Currently, available CAR-T cell therapies are customized for individual patients. They are created by harvesting T cells from a patient and reengineering them in the laboratory to express the CAR on their surfaces. Terminal differentiation may progress as a patient’s T cells are exposed to the tumor microenvironment and acquire a senescent and exhausted phenotype [95]. The percentage of CD4+ and CD8+ CAR-T cells expressing the senescent surface markers CD57 and KLRG-1 increased after infusion; however, the degree of increase was greater in CD8+ T cells than in CD4+ cells [96,97]. In activated human T-cells, CD57 is upregulated upon contact with CD57+ target cells [97]. CD57 expression may not necessarily reflect T-cell senescence or terminal T-cell differentiation state. However, the detailed mechanism and its direct relationship with the therapeutic efficacy remain unclear and require further investigation. Additionally, pretreatment with DNA/RNA synthesis inhibitors, such as DXR, results in inadequate amounts of CAR-T cells or low-quality CAR-T cells that can be recovered [98]. Clinical data suggest that pretreatment with cyclophosphamide and cytarabine selectively reduces early lineage T-cells associated with the proliferation of productive CAR-T cells [99]. These chemotherapeutic agents have been reported to induce T-cell senescence [19]. The inhibition of p16INK4a/p38MAPK signaling after T cell stimulation could limit the development of senescence features and increase the proportion of cytokine-secreting CAR-T cells [39,100]. These results suggest that TIS-T cells also affect CAR T-cell therapy.

**Table 1 cancers-15-05835-t001:** Positive or Negative influence of senescent T cells.

Influence of Senescent T Cells	Key Results	References
Gain of innate-like functions	Elevated the expression of NK-like receptors	[72]
	KIR/NKG2A+ T cells exert cytolytic activity in a TCR-independent manner	[73]
Suppression of anti-tumor immunity	Tumor-infiltrating T cells exhibit senescent characteristics at tumor sites	[11,72,73,74]
Prognostic impact	The frequency of senescent T cells in patients with AML in CR is lower than the frequency of senescent T cells in patients with untreated AML	[14]
	Patients with stage IV lung cancer have an increased number of senescent T cells	[82]
	High levels of senescent T cells in the peripheral blood are associated with poor prognosis	[13,75,83,84]
Effects of ICIs	Senescent T cells is associated with resistance to checkpoint inhibitor blockade in patients with melanoma	[87]
CAR-T cell therapies	The percentages of CD4+ and CD8+ CAR-T cells expressing the senescent surface markers CD57 and KLRG-1 increased after infusion	[96,97]
	Pre-treatment with DNA/RNA synthesis inhibitors such as DXR results in inadequate amounts of CAR-T cells or low-quality CAR-T cells that can be recovered	[98]

NK, natural killer; KIR, killer cell Ig-like receptor; TCR, T cell receptor; AML, acute myeloid leukemia; CR, complete remission; ICIs, immune checkpoint inhibitors; CAR-T cell, chimeric antigen receptor T cell; KLRG-1, killer cell lectin-like receptor subfamily G member 1.

## 9. Potential Therapies Targeting Senescent T Cells

Polyphenols, probiotics, and metabolites have been reported to eliminate senescent T cells or reverse the senescent phenotype. Shrma et al. observed that treatment of aged Swiss albino mice with epigallocatechin-3-gallate increased the percentage of CD3+/CD8+ T cells in splenocytes and CD28 expression in PBMNCs [101]. Syringaresinol treatment of middle-aged mice increased the total number of CD3+ T cells and naive T cells and delayed immune senescence [102]. Lactobacillus fermentation products also adequately rescued cells from stress-induced senescence, as evidenced by the reduced p16Ink4a/p53/p21WAF1 expression [103]. Miyazaki et al. observed that ingestion of heat-sterilized Lactobacillus gasseri increased the number of CD8+ T cells and reduced the loss of CD28 expression in elderly patients. Nicotinamide adenine dinucleotide (NAD+) is a metabolite involved in numerous redox reactions that fuels diverse metabolic pathways. Treatment with nicotinamide riboside, a precursor of NAD+, improves mitochondrial fitness in both senescent and exhausted CD8+ T cells [104]. In addition, the treatment of primary human CD4 + T cells expressing the senescence-inducing human T-cell leukemia virus type 1 Tax oncoprotein with the ROS scavenger N-acetyl cysteine suppressed the expression of senescence markers [105]. Understanding the interplay between other factors known to induce T-cell senescence, such as oxidative stress, DNA damage, and chronic inflammation, may provide valuable insights into the development of therapeutic strategies to relieve T-cell senescence and enhance immune function.

Melanoma and colon tumor growth were inhibited in mice treated with nicotinamide riboside and an anti-PD-1 antibody, and the combination of nicotinamide riboside supplementation and ICI therapy elicited additive antitumor effects [104]. The transduction of CD27 [106] or CD28 [107], for which expression is lost in senescent cells, enhances the survival and anti-tumor effects of CAR-T cells. Moreover, growing evidence demonstrates that IL-7, IL-15, and IL-21 are associated with delayed or reversed senescence of antigen-specific CD8+ T cells and CAR-T cells [108,109,110]. IL-7 prevents the development of a senescent phenotype by sustaining the expression of CD27/CD28 and maintaining the functionality of T cells with suppressor functions [108]. Furthermore, IL-15 and IL-21 prevent terminal differentiation of tumor antigen-specific T cells and promote their expansion and effector functions [109,110]. These reports suggest that the removal of senescent T cells is a key factor in the efficacy of cancer immunotherapy.

## 10. Conclusions

T-cell senescence is induced not only by aging and repeated antigen stimulation but also by treatment with DNA-damaging chemotherapeutic agents. Although the characteristics of TIS-T cells and general senescent T cells are similar, TIS-T cells exhibit enhanced PD-1 expression. Senescent T cells acquire innate-like functions by expressing NK cell-associated receptors and secreting SASP, including inflammatory cytokines. However, the T-cell senescence-associated loss of proliferative capacity and decreased expression of co-stimulatory molecules cause a decrease in T-cell immunocompetence. Overall, T-cell senescence causes negative effects, such as the suppression of anti-tumor immunity. Additionally, senescent T cells have been suggested to be involved in immune checkpoint inhibition and CAR-T cell therapy, both of which have been the focus of attention in recent years. These treatments are often preceded by intense chemotherapy, which may affect the TIS-T cells induced by antitumor drugs. Regulation of senescent T cells may also be important for optimizing the efficacy of immunotherapy. Future research to understand the interactions between cancer therapies and other factors known to induce T-cell senescence, such as oxidative stress, DNA damage, epigenetics, and metabolomics, could provide valuable insights into the development of therapeutic strategies to mitigate T-cell senescence and enhance immune function.

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
