# Peer review of "Implications of Senescent T Cells for Cancer Immunotherapy"

_cancers, 2023, doi:10.3390/cancers15245835_

Round 1

Reviewer 1 Report

Comments and Suggestions for Authors

The review "Therapy-induced cellular senescence in T cells" provides valuable insights into the induction and characteristics of senescence in T cells, focusing on the effects of chemotherapy and potential implications for cancer immunotherapy. Here are some comments and suggestions for the author.

1. The title could be more descriptive and specific to help readers better understand the focus of the review. For example, consider a title like "Implications of Therapy-Induced Senescence in T Cells for Cancer Immunotherapy."

2. The abstract is well-structured and informative, providing an overview of the study's objectives and key findings. However, it can be made more concise. For example, you can combine sentences like "In this review, we discuss the characteristics of senescent T cells, including therapy-induced senescent T cells" with the preceding sentence. 

3. The introduction provides a clear background on T cells and their role in cancer immunity, which is great. It would be good to provide a brief background on cellular senescence and its relevance in the context of cancer immunotherapy. This would help readers who may not be familiar with the topic. When introducing T cells, explain what TCRs (T cell receptors) are and their role in antigen recognition. This would provide a more comprehensive understanding for readers.

4. The paper is generally well organized. However, consider using subheadings within the review sections to make it easier for readers to navigate and locate specific information. For example, separate sections for "Characteristics of Senescent T Cells" and "Therapy-Induced Senescent T Cells" would help in organizing the content.

5. The conclusion is concise and provides a good summary of the key points discussed in the review. However, you can emphasize the potential future directions in research related to T cell senescence and its implications for cancer therapy.

Comments on the Quality of English Language

The review is generally well-written, but some sentences are quite long and complex. Breaking them into shorter sentences for easier readability might improve the overall flow.

Define acronyms and technical terms upon first use to ensure clarity for readers who may not be experts in the field.

Author Response

1. The title could be more descriptive and specific to help readers better understand the focus of the review. For example, consider a title like "Implications of Therapy-Induced Senescence in T Cells for Cancer Immunotherapy."

Reply

We have changed the title to according to your kind suggestion.

2. The abstract is well-structured and informative, providing an overview of the study's objectives and key findings. However, it can be made more concise. For example, you can combine sentences like "In this review, we discuss the characteristics of senescent T cells, including therapy-induced senescent T cells" with the preceding sentence.

Reply

I have revised the sentence in “abstract” according to your kind suggestion.(lime 28 - 29)

3. The introduction provides a clear background on T cells and their role in cancer immunity, which is great. It would be good to provide a brief background on cellular senescence and its relevance in the context of cancer immunotherapy. This would help readers who may not be familiar with the topic. When introducing T cells, explain what TCRs (T cell receptors) are and their role in antigen recognition. This would provide a more comprehensive understanding for readers.

Reply

A brief background on cellular senescence and its relevance to cancer immunotherapy has been added to the "Introduction" section. In addition, I have added a description of the TCRs in the section describing T cells.(Line 50 - 62)

4. The paper is generally well organized. However, consider using subheadings within the review sections to make it easier for readers to navigate and locate specific information. For example, separate sections for "Characteristics of Senescent T Cells" and "Therapy-Induced Senescent T Cells" would help in organizing the content.

Reply

I added subheadings to each section (including Functional markers, Cell surface markers, Senescence-associated secretory phenotype and Metabolic features) of the paper and separated sections for "Characteristics of Senescent T Cells" and "Therapy-Induced Senescent T Cells".

5. The conclusion is concise and provides a good summary of the key points discussed in the review. However, you can emphasize the potential future directions in research related to T cell senescence and its implications for cancer therapy.

Reply

I added a new section on "Potential Therapies Targeting Aging T Cells," highlighting future directions in research on T cell aging and its impact on cancer therapy.(Line 402 - 433)

 Comments on the Quality of English Language

The review is generally well-written, but some sentences are quite long and complex. Breaking them into shorter sentences for easier readability might improve the overall flow.

Define acronyms and technical terms upon first use to ensure clarity for readers who may not be experts in the field.

Reply

The manuscript was proofread in English, including the added sections, and some changes were made throughout the text.

Reviewer 2 Report

Comments and Suggestions for Authors

Tetsuhiro Kasamatsu reviewed therapy-induced cellular senescence in T cells. This study is interesting and give insight into senescence of T cells, and provide knowledge of cancer immunotherapy. I have some concerns:

1.     Although this manuscript is well prepared, the language should be polished by a native speaker at deep extent.

2.     The title should be modified, from 1 to 7.3 the author mainly introduced general function or phonotype of senescent T cells, while therapy-related T cells senescence, or role of T cells senescence in cancer therapy account for little part of the manuscript.

3.     Some part of the review should be clarified more clearly and deeply. For example, the terminally differentiated phenotype and the terminal-differentiation markers expression, the function change and their contribute to cancer progression and cancer treatment should be further discussed.

4.     Mechanism underlying senescent T cells phenotypes should be discussed, for example, the tumor microenvironment, some epigenetics and metabiotic events all contribute to the phenotype and function change of senescence cells.

5.     Some factors such as polyphenols, probiotic microbes, and some metabiotic products were reported reverse immunosenescence, possible therapeutic strategies for preventing or rejuvenating senescence in tumour-specific T cell should be mentioned, whether they were reported to increase cancer treatment effect and their potential should be further discussed in a separated part.

6.     Some great reviews about T cell senescence, immunotherapy should be referred and cited.

7.     Single-cell RNA sequencing (scRNA-seq) could provide in-depth analysis of cell heterogeneity, which has also been used in the analysis of T cell senescence, besides classical marker the author mentioned in the article, novel molecular hallmarks of senescent T-cells should be briefly summarized in the reviewed.

Comments on the Quality of English Language

Minor editing required.

Author Response

List of Corrections

To the comments from Referee

1. Although this manuscript is well prepared, the language should be polished by a native speaker at deep extent.

Reply

The manuscript was proofread again by a native English speaker.

2. The title should be modified, from 1 to 7.3 the author mainly introduced general function or phonotype of senescent T cells, while therapy-related T cells senescence, or role of T cells senescence in cancer therapy account for little part of the manuscript.

Reply

The title has been changed because several reviewers pointed out.

3. Some part of the review should be clarified more clearly and deeply. For example, the terminally differentiated phenotype and the terminal-differentiation markers expression, the function change and their contribute to cancer progression and cancer treatment should be further discussed.

Reply

A new section on “Differentiation phenotypes” was added to the manuscript. In that section, we discuss terminal differentiation phenotypes and how they contribute to cancer.(Line 235 - 268)

4. Mechanism underlying senescent T cells phenotypes should be discussed, for example, the tumor microenvironment, some epigenetics and metabiotic events all contribute to the phenotype and function change of senescence cells.

Reply

The effect of epigenetic events on T cell terminal differentiation was added to the "Differentiation phenotypes " section. (Line 257 - 268)

5. Some factors such as polyphenols, probiotic microbes, and some metabiotic products were reported reverse immunosenescence, possible therapeutic strategies for preventing or rejuvenating senescence in tumour-specific T cell should be mentioned, whether they were reported to increase cancer treatment effect and their potential should be further discussed in a separated part.

Reply

A new section "Potential therapies targeting senescent T cells" has been added to address potential therapeutic strategies targeting senescent T cells. (Line 402 - 433)  In that section, I describe factors that reverse T cell senescence, such as polyphenols and probiotic microorganisms. (Line 403 - 415)

6. Some great reviews about T cell senescence, immunotherapy should be referred and cited.

Reply

In a new section of the manuscript added in response to reviewers' suggestions, we cited several additional great reviews about T cell senescence.

7. Single-cell RNA sequencing (scRNA-seq) could provide in-depth analysis of cell heterogeneity, which has also been used in the analysis of T cell senescence, besides classical marker the author mentioned in the article, novel molecular hallmarks of senescent T-cells should be briefly summarized in the reviewed.

Reply

I have added a description of novel molecular markers by single-cell RNA sequencing to the "Functional markers" section.(Line 121 - 128)

Reviewer 3 Report

Comments and Suggestions for Authors

This review is on therapy-induced cellular senescence in T cells. It discusses the characteristics of senescent T cells, including those induced by therapy, and explores the positive and negative effects of therapy-induced cellular senescence on T cells. The review highlights that senescent T cells are defective in proliferation and effector function and can accumulate due to aging, chronic viral infections, and autoimmune diseases. The author claim that certain chemotherapeutic agents can induce senescence in cancer cells and also in T cells. The review concludes that T-cell senescence has negative effects on T cell immunocompetence and can suppress anti-tumor immunity. Additionally, it discusses the implications of therapy-induced senescence for immune checkpoint inhibitors and CAR-T cell therapies.

Based on the provided context, I would suggest the authors of the review to include a paragraph on T cell senescence after oxidative stress or DNA damage. They should also expand on the future perspectives in this area.

In the context of T cells, oxidative stress and DNA damage can contribute to the accumulation of senescent T cells. Some therapies can increase the presence of ROS and consequently enhance the onset of cellular senescent in T cells (PMID: 20219913; PMID: 27547291; PMID: 33868291).

Future studies could investigate the involvement of specific DNA repair mechanisms, and the impact of antioxidant defense systems on T cell senescence. Additionally, understanding the interplay between oxidative stress, DNA damage, and other factors known to induce T cell senescence, such as chronic viral infections and autoimmune diseases, could provide valuable insights into the development of therapeutic strategies to mitigate T cell senescence and enhance immune function.

Author Response

List of Corrections

To the comments from Referee

Based on the provided context, I would suggest the authors of the review to include a paragraph on T cell senescence after oxidative stress or DNA damage. They should also expand on the future perspectives in this area.

In the context of T cells, oxidative stress and DNA damage can contribute to the accumulation of senescent T cells. Some therapies can increase the presence of ROS and consequently enhance the onset of cellular senescent in T cells (PMID: 20219913; PMID: 27547291; PMID: 33868291).

Future studies could investigate the involvement of specific DNA repair mechanisms, and the impact of antioxidant defense systems on T cell senescence. Additionally, understanding the interplay between oxidative stress, DNA damage, and other factors known to induce T cell senescence, such as chronic viral infections and autoimmune diseases, could provide valuable insights into the development of therapeutic strategies to mitigate T cell senescence and enhance immune function.

Reply

I appreciate your comments and suggestions.

According to your kind suggestion, I have added the description of the involvement of oxidative stress and DNA damage in T cell senescence in the "Metabolic features" section. (Line 296 - 309)

I added a new section on "Potential Therapies Targeting Aging T Cells," highlighting future directions in research on T cell aging and its impact on cancer therapy.(Line 402 - 433)

In addition, The following sentence was also added to the Conclusion as a future perspective, ” Future research to understand the interactions between cancer therapies and other factors known to induce T cell senescence, such as oxidative stress, DNA damage, epi-genetics, and metabolomics, could provide valuable insights into the development of therapeutic strategies to mitigate T cell senescence and enhance immune function.”.(Line 447 - 451)

Reviewer 4 Report

Comments and Suggestions for Authors

This is an interesting review. I enjoyed reading it. The senescent field is vast, and it is nice to read a focussed review on T cell senescence.

I do not have any major comments, however, I think the review would benefit from a more elaborated discussion on the difference sand similarities between T cell senescence and T cell exhaustion. The irreversibility of senescence in the cancer context is being questioned. At the moment it is not clear to me whether the differences are as clear as stated in the review.

The author could for example expand the statements about this subject (T cell senescence and exhaustion), bringing up the critical experiments that were carried out in the references cited.  I suggest the authors add a new heading:" T cell senescence and exhaustion" or something like this, to present and discuss the critical data in a more detailed manner.

I think this is important because although in lanes 116-126, it is concluded the differences are clear, the author states the opposite in lines 240-241. 

Minor comments:

1. Line 92. Two dots between the reference 21.

2. Lines 259 and 260. I do not understand the sentence.

Comments on the Quality of English Language

I think the English is good in general.

Author Response

List of Corrections

To the comments from Referee

This is an interesting review. I enjoyed reading it. The senescent field is vast, and it is nice to read a focussed review on T cell senescence.

I do not have any major comments, however, I think the review would benefit from a more elaborated discussion on the difference sand similarities between T cell senescence and T cell exhaustion. The irreversibility of senescence in the cancer context is being questioned. At the moment it is not clear to me whether the differences are as clear as stated in the review.

The author could for example expand the statements about this subject (T cell senescence and exhaustion), bringing up the critical experiments that were carried out in the references cited.  I suggest the authors add a new heading:" T cell senescence and exhaustion" or something like this, to present and discuss the critical data in a more detailed manner.

I think this is important because although in lanes 116-126, it is concluded the differences are clear, the author states the opposite in lines 240-241.

Reply

I appreciate your comments and suggestions.

I added a new subsection "Senescent T cells and exhausted T cells " to section " Cell surface markers ". I have added the description of the differences and similarities between T cell senescence and T cell exhaustion in the new sub-section. (Line 175 - 197)

I have also avoided explicit statements about the irreversibility of senescent T cells and added a new section “Potential therapies targeting senescent T cells” on their prospects as a therapeutic target along with the possibility of reversibility. (Line 402 - 433)

Minor comments:

  1. Line 92. Two dots between the reference 21.

Reply

Text has been revised.

  1. Lines 259 and 260. I do not understand the sentence.

Reply

Since CD57+ target cells upregulate CD57 expression on CAR-T cells, CD57 expression may not necessarily reflect T cell senescence or terminal T cell differentiation state. I added the sentence " CD57 expression may not necessarily reflect T cell senescence or terminal T cell differentiation state." after that one. (Line 387 - 388)

Round 2

Reviewer 2 Report

Comments and Suggestions for Authors

No further suggesstions

Reviewer 3 Report

Comments and Suggestions for Authors

none